# Therapeutic Use of the Antimicrobial Peptide PNR20 to Resolve Disseminated Candidiasis in a Murine Model

**DOI:** 10.3390/jof9121149

**Published:** 2023-11-28

**Authors:** Jeisson Micelly-Moreno, Adriana Barreto-Santamaría, Gabriela Arévalo-Pinzón, Carolina Firacative, Beatriz L. Gómez, Patricia Escandón, Manuel A. Patarroyo, Julián E. Muñoz

**Affiliations:** 1Faculty of Health Sciences, Universidad Colegio Mayor de Cundinamarca, Bogota 110311, Colombia; jmicelly@unicolmayor.edu.co; 2Studies in Translational Microbiology and Emerging Diseases (MICROS) Research Group, School of Medicine and Health Sciences, Universidad de Rosario, Bogota 111221, Colombia; cfiracative@gmail.com (C.F.); beatriz.gomez@urosario.edu.co (B.L.G.); 3Receptor-Ligand Department, Fundación Instituto de Inmunología de Colombia (FIDIC), Bogota 111321, Colombia; adrianasantamaria10@gmail.com; 4Microbiology Department, Faculty of Sciences, Pontificia Universidad Javeriana, Carrera 7 #40–62, Bogota 110231, Colombia; arevalog@javeriana.edu.co; 5Microbiology Group, Instituto Nacional de Salud, Bogota 111321, Colombia; pescandon@ins.gov.co; 6Molecular Biology and Immunology Department, Fundación Instituto de Inmunología de Colombia (FIDIC), Bogota 111321, Colombia; mapatarr.fidic@gmail.com; 7Microbiology Department, Faculty of Medicine, Universidad Nacional de Colombia, Bogota 111321, Colombia; 8Public Health Research Group, School of Medicine and Health Sciences, Universidad del Rosario, Bogota 111221, Colombia

**Keywords:** *Candida auris*, *Candida* spp., antimicrobial peptides, resistance, candidiasis

## Abstract

Invasive fungal infections (IFIs) caused by *Candida* species are an emerging threat globally, given that patients at-risk and antifungal resistance are increasing. Antimicrobial peptides (AMPs) have shown good therapeutic capacity against different multidrug-resistant (MDR) microorganisms. This study evaluated the activity of the synthetic peptide, PNR20, against *Candida albicans* ATCC 10231 and a MDR Colombian clinical isolate of *Candida auris*. Perturbation of yeast cell surface was evaluated using scanning electron microscopy. Cell viability of Vero cells was determined to assess peptide toxicity. Additionally, survival, fungal burden, and histopathology of BALB/c mice infected intravenously with each *Candida* species and treated with PNR20 were analyzed. Morphological alterations were identified in both species, demonstrating the antifungal effect of PNR20. In vitro, Vero cells’ viability was not affected by PNR20. All mice infected with either *C. albicans* or *C. auris* and treated with PNR20 survived and had a significant reduction in the fungal burden in the kidney compared to the control group. The histopathological analysis in mice infected and treated with PNR20 showed more preserved tissues, without the presence of yeast, compared to the control groups. This work shows that the utilization of PNR20 is a promising therapeutic alternative against disseminated candidiasis.

## 1. Introduction

The significant increase in the number of fungal infections, including invasive candidiasis, raises alarm due to the high morbidity and mortality rates of patients around the world [1]. Incidence and spectrum of invasive fungal infections (IFI) are constantly rising, even considering that advances in medical care and life-saving treatments contribute to a growing number of patients at-risk [2]. Globally, invasive candidiasis is the most frequent mycosis, causing nearly 750,000 cases per year, with a mortality rate close to 40% [2,3]. In Colombia specifically, this mycosis accounts for about 75% of all cases of IFI occurring in hospitalized patients, mostly those with medical interventions and iatrogenic conditions [4].

Among *Candida* species, *Candida albicans* heads the list of infectious agents associated with health care, causing predominantly bloodstream infections [1,3]. However, other species, such as *Candida parapsilosis*, *Candida glabrata*, *Candida tropicalis*, *Candida krusei*, and *Candida auris*, have been increasing in incidence, which is of concern when selecting treatment, as many non-*albicans* species have reduced susceptibility or are intrinsically resistant to one or more commonly used antifungal drugs [5,6]. *C. auris*, in particular, is considered an emergent multidrug-resistant species that has already spread to all continents as an important intrahospital pathogen [6]. Characterized by its capability to persist for prolonged periods both on patients’ skin and in environmental surfaces, *C. auris* can be shed in healthcare settings [7]. In addition, its adaptability to various abiotic factors, such as pH and toxic metabolites, allows *C. auris* to evade the immune system and tolerate stress responses, which makes it a pathogen of great interest [8,9,10].

In Colombia, *C. auris* was notified for the first time in 2016 by the National Institute of Health (INS) [11]. Since then, the epidemiological alert has been under constant observation in the country as internationally [12]. The first report of this pathogen informed of 17 cases from six institutions in the northern region of Colombia, with a 30-day mortality rate of 35.2% [13]. However, together with the Centers for Disease Control and Prevention (CDC), a later study carried out by the INS and hospitals in three cities in Colombia, investigated acute outbreaks of candidiasis in geriatric and pediatric patients, during 2015–2016, finding a total of 40 cases of infection by *C. auris* with a 30-day mortality of 43% [14]. These reports, together with others from Colombia and the region, highlight the need for constant surveillance of this pathogen [15,16,17].

Microbial resistance, not only in *C. auris* but also in other fungal species, is another public health concern worldwide, given that the number of cases of microorganisms that are resistant to the different available antimicrobials is constantly increasing; hence, therapeutic options are scarcer. Therefore, there is a need to search for and develop new therapeutic agents with different mechanisms of action to help fight against commonly occurring pathogens [18]. Compared to antibiotics, there are clear limitations regarding the number of medications suitable for treating fungal infections. Antifungals, in addition to being a small group, are becoming less effective every year due to the increase in resistance [19]. This fact prompts the completion of research to develop viable, sustainable, economical, and non-toxic therapeutic alternatives, especially against microorganisms of clinical importance that have high mortality rates, such as *C. auris* [7,13].

Antimicrobial peptides (AMPs) have been increasingly studied as a therapeutic alternative against multidrug-resistant microorganisms. First, because resistance to these molecules is rare, despite prolonged exposure to them [20]. In addition, modifications of the structure and sequence of amino acids can be made to enhance their antimicrobial effect, as well as to reduce possible cytotoxicity, making them a line of interest in the short and long term [21]. AMPs have characteristics that allow them to interact with the membranes of the target microorganisms, such as amphipathicity, positive charge, helical structures, and hydrophobic amino acid content, that give them an active potential against negatively charged cell membranes, triggering a membranolytic effect [22].

PNR20 (1609), a 20-amino-acid peptide with a cationic charge, has demonstrated in vitro activity against clinically important gram-negative and gram-positive bacteria, such as *Escherichia coli*, *Pseudomonas aeruginosa*, *Staphylococcus aureus*, and *Enterococcus faecalis* (unpublished results). Antifungal activity of this peptide has recently been reported against *Candida* species in vitro, including *C. albicans* and *C. auris* [23]. The objective of this work was, therefore, to evaluate the antifungal activity of the synthetic peptide PNR20 in vivo in a murine model of disseminated candidiasis, as well as to better understand the mechanism of action of this AMP against *C. albicans* and a multidrug-resistant clinical isolate of *C. auris*.

## 2. Materials and Methods

### 2.1. Synthesis and Purification of the Antimicrobial Peptide PNR20

PNR20 was designed using random selection of 10 polar and 10 non-polar amino acids at the Fundación Instituto de Inmunología de Colombia (FIDIC) in Bogotá, as indicated previously [23]. After design, PNR20 was synthesized commercially with 95% purity, by Peptide 2.0 (Chantilly, VA, USA). High-performance liquid chromatography (HPLC) and mass spectrometry (MS) were used for peptide purification and identification.

### 2.2. Candida Strains

For all experiments, the reference strain of *Candida albicans* ATCC 10231 and the multidrug-resistant Colombian clinical isolate of *Candida auris* H0059-13-2251, provided by the National Institute of Health (INS), were used. Considering that the antifungal susceptibility profile of the *C. auris* isolate is a minimum inhibitory concentration (MIC) of 64 µg/mL for fluconazole and a MIC of 8 µg/mL for amphotericin B, this isolate is considered resistant to both antifungals [24]. All strains were preserved in 10% glycerol at −80 °C. Three days before the experiments began, each strain was subcultured on a Sabouraud dextrose agar (Becton, Dickinson, New Jersey, NJ, USA) and incubated at 37 °C for 24 h, in order to recover exponentially growing yeasts.

### 2.3. Scanning Electron Microscopy (SEM)

The morphology of *C. albicans* ATCC 10231 and *C. auris* H0059-13-2251 after treatment with PNR20 was evaluated using scanning electron microscopy (SEM) and focused ion beam FE-MEB LYRA3 of TESCAN (Brno, Czech Republic). A cell suspension of *C. albicans* and *C. auris* was adjusted, separately, to 1.0–5.0 × 10^6^ colony-forming units (CFU)/mL, which equals the 0.5 McFarland standard. Each suspension was mixed with the antimicrobial peptide to obtain a final concentration of 6.25 µM for *C. albicans* and 50 µM for *C. auris* and incubated at 37 °C for 3 h. These values correspond to one dilution smaller than the MIC of the peptide per species, as established previously [23]. Subsequently, the yeasts were centrifuged and fixed in 2.5% glutaraldehyde overnight in the refrigerator. Cells were washed three times with milli-Q ultrapure water and dehydrated with serial dilutions of ethanol (from 50 to 100%). Finally, the cells were observed in a scanning electron microscope (Dual Beam SEM-FIB), at the MicroCore microscopy center of the Universidad de los Andes, in Bogotá. Yeasts without antimicrobial peptide treatment were used as a control.

### 2.4. In Vitro Cytotoxicity Assay with the Vero Cell Line

The cytotoxicity of PNR20 was evaluated in the Vero cell line. Vero cells were seeded in 96-well plates at a concentration of 5 × 10^3^ cells/well resuspended in Dulbecco’s Modified Eagle Medium (DMEM) medium (Gibco, Massachusetts, MA, USA) enriched with 1 g/L of D-glucose, L-Glutamine and 110 mg/L of sodium pyruvate. The cells were incubated at 37 °C in 5% CO_2_ for 24 h, and, subsequently, different concentrations of PNR20, ranging from 3.125 μM to 200 μM, were added to the wells and incubated for 24 h. The toxicity of PNR20 was determined by the cell viability of Vero cells, which was measured using the MTT (Thiazolyl Blue Tetrazolium Bromide: Sigma, St Louis, MO, USA) method according to the protocol of Danihelová et al. [25]. The controls used for this experiment were growth control (untreated cells) and death control (cells treated with 100% dimethyl sulfoxide (DMSO), Sigma-Aldrich, Missouri, MO, USA). Per experiment, three replicates were carried out.

### 2.5. Murine Model of Disseminated Candidiasis

Female BALB/c mice aged between 6 and 8 weeks and weighing approximately 25–30 g were raised and maintained in the Animal Facility of the Faculty of Veterinary Medicine and Zootechnics of Universidad Nacional de Colombia, in Bogotá. All experiments that involved the use of animals were approved by the bioethics committee of the Universidad Nacional de Colombia (Protocol number CB-FMVZ-UN-039-2021) and were carried out following international recommendations. To resemble a model of disseminated candidiasis that does not resolve spontaneously, four groups of 16 mice each were immunosuppressed with two doses of cyclophosphamide (Sigma, Aldrich, Missouri, MO, USA) four days and one day before infection with the *Candida* species, at a dose of 100 mg/Kg administered intraperitoneally (IP). For infection, and due to immunosuppression, 1 × 10^3^ cells of *C. albicans* ATCC 10231 and 1 × 10^3^ cells of *C. auris* H0059-13-2251 were inoculated intravenously. After infection, the immunosuppressant was administered in the aforementioned dose, every four days until the end of the experiment, following the protocol described by Rossi and collaborators in 2012 [26]. One day after infection and for seven days following, a group of 12 mice was treated intraperitoneally, with 3 mg/Kg/day of PNR20. Another group of 12 mice was also treated intraperitoneally, for 7 days, with 20 mg/Kg/day of fluconazole. A third group of 12 mice was treated with both PNR20 and fluconazole (3 mg/Kg/day and 20 mg/Kg/day, respectively). Finally, a fourth group of 12 mice was kept immunosuppressed and untreated but inoculated intraperitoneally with PBS instead, for the same number of days as treated mice. The dose of the peptide used for treatment was chosen considering previous studies assessing AMPs to treat experimental candidiasis [26], as well as considering the molecular weight and the MIC of the peptide, as established previously [23]. The dose of fluconazole was as well-established elsewhere [26].

For surveillance analysis, 10 mice per group were observed daily for signs of infection (e.g., difficult breathing, ruffled fur, lethargy, poor eating) and deaths were recorded for 40 days. For fungal-burden determination, six mice per group were euthanized using a rapid, painless, stress-free death with CO_2_, on the eighth day of infection. After death, the kidneys, spleen, and liver of all mice were collected, weighed, macerated, and mixed with 1 to 2 mL of PBS, depending on the size. From the suspension of each organ, 100 µL were plated on Sabouraud dextrose agar to determine the fungal load. For organs from mice infected by *C. auris*, an additional 1:10 dilution was done before platting. For histopathological analysis, one of the kidneys from the animals belonging to each group was collected and placed in a tube with 10% formalin (Sigma, Aldrich, Missouri, MO, USA), to later include the organ in paraffin blocks and thus perform the 5 µm histological sections on a microtome (Microm HM 325, Thermo Fisher Scientific Inc., Waltham, MA, USA). The slides obtained were stained using Periodic Acid-Schiff (PAS), and were observed under an optical microscope (Zeiss, Axio Lab 5. ICS System (Infinity Color-Corrected System, Carl Zeiss Microscopy, New York, NY, USA). Histological analysis of spleens and livers were not carried out, considering the low fungal burdens in these organs.

### 2.6. Statistical Analysis

The statistical comparisons were performed using analysis of variance (Kruskal–Wallis test) and one-way ANOVA. Survival curves were constructed using the Kaplan–Meier method, and the curves were compared using the Log Rank test (Mantel–Cox). Statistical analyses and graphs were conducted using GraphPad Prism version 8.0 (GraphPad Software, San Diego, CA, USA). A *p*-value less than 0.05 was considered statistically significant.

## 3. Results

### 3.1. PNR20 Alters the Morphology of C. albicans and C. auris

PNR20 had the ability to alter the cell structure of the reference strain of *C. albicans* ATCC 10231, compared to the control (untreated yeast). Changes were observed at the cell surface level, as, likely, the cell membrane was damaged. Yeasts treated with 6.25 μM of PNR20 presented alterations such as collapse of the cell surface and the appearance of the yeast is wrinkled compared to the untreated control (Figure 1).

With respect to the multidrug-resistant *C. auris* H0059-13-2251, PNR20 also showed potent antifungal activity. After exposure of this yeast to 50 µM of the peptide, damage to the cell surface and severe cell morphological alterations were observed (Figure 2).

### 3.2. PNR20 Does Not Induce Toxicity in Vero Cells

PNR20 is safe to use with mammalian cells, since the peptide did not induce cytotoxicity in Vero cells treated for 24 h with concentrations ranging from 3.125 µM to 200 µM. Even though there was a slight reduction in cell viability compared to untreated cells, this was not statistically significant (*p*-value > 0.05) (Figure 3). The higher concentration of PNR20 (200 µM) reduced Vero cells viability by only 15.6%.

### 3.3. PNR20 Resolves Murine Disseminated Candidiasis Caused by Both C. albicans and C. auris

Ten BALB/c mice per group were inoculated intravenously with 1 × 10^3^ yeast cell of *C. albicans* or *C. auris* and observed once daily for 40 days to determine mortality. Disseminated infection with *C. albicans* without treatment resulted in 100% mortality at 28 days post-infection (Figure 4), concomitant with clinical signs of disease, such as lethargy or somnolence and weight loss. Similar mortality was observed after *C. auris* infection without treatment, since all mice died after 30 days. When treated for 7 days, either with PNR20 alone or in combination with fluconazole, all mice, infected by either *C. albicans* or *C. auris* survived (Figure 4). PNR20 monotherapy or in combination with FCZ was better than the treatment with fluconazole alone. Mean survival time in *C. albicans* is 21 days for the untreated control, and mean survival time in *C. auris* is 27 days for untreated control, and 29 days for treatment with fluconazole. 

Regarding the fungal burden after *Candida* infection and 7 days of treatment, the reduction of the number of yeast cells in the kidneys of mice infected with *C. albicans* was evident to a large extent when treating the animals with PNR20 alone or in combination with fluconazole (Figure 5A). It should be noted that the efficiency of PNR20 in killing *C. albicans* in kidneys is comparable to that of fluconazole, since the average fungal load obtained in the kidneys was reduced by 80% after monotherapy with either PNR20 or FCZ. Notably, combination therapy resulted in a fungal-load reduction close to 95%, showing the synergisms between the peptide and the azole.

Similar findings on the fungal load were noticed in mice infected with *C. auris* and treated for 7 days with PNR20 alone, since the number of yeast cells in their kidneys decreased significantly compared to untreated mice (Figure 5B). However, when the mice infected with *C. auris* were treated with PNR20 plus FCZ, or FCZ alone, the renal fungal load did not exhibit a significant reduction.

Interestingly, even though the infection of the mice was carried out with the same inoculum, *C. auris* infection propagates much more than *C. albicans*, as the fungal burden in kidneys of mice infected with *C. auris* is 100 times higher compared to the kidneys of mice infected with *C. albicans*. Fungal burdens in spleens and livers of animals treated with either *C. albicans* or *C. auris* were significantly low; therefore, these results are not shown.

A histological analysis revealed that, compared with uninfected mice, the kidney tissue of infected but untreated animals presented severe organ involvement and several foci of *C. auris* (Figure 6A,B). The tissue of the mice treated with PNR20 (Figure 6C) shows a decrease in the fungal load. However, it is important to highlight that in the organs of the treated animals a considerable amount of cellular infiltrate could be observed, which may suggest a possible immunomodulatory activity of the AMP PNR20. Figure 6D shows the kidney tissue of mice infected and treated with 20 mg/Kg/day for 7 days of fluconazole. Even though *C. auris* H0059-13-2251 is resistant to this antifungal, a slight decrease in the fungal load is observed in the organs of these animals. This result correlates with what was observed in the fungal-load analysis through CFUs (Figure 5B). Very similar results were observed in the kidneys of mice infected with *C. albicans* ATCC 10231, in which no hyphae or pseudohyphae formation was observed. Due to the significantly low fungal burdens in spleens and livers of animals with disseminated candidiasis by *C. auris* or *C. albicans*, histological analyses of these organs were not carried out.

## 4. Discussion

The increase in the number of cases of infection caused by *Candida* species worldwide, including *C. auris*, is alarming. An epidemiological study carried out in seven Latin American countries, including Colombia, reported an incidence of candidemia of 1.18 cases per 1000 admitted to hospitals. Of this incidence, Colombia registers one of the highest, presenting 1.98 cases per 1000 admissions, an alarming figure compared to the incidence reported in the United States and Europe, which is below 1 case per 1000 admissions [27]. In addition to the high incidence of candidemia in the country, *C. auris* infections in Colombia continue to be a priority, as this pathogen is exceeding the number of cases caused by *C. albicans*, and is becoming increasingly common as a microorganism affecting patients in intensive care units with unacceptably high mortality rates [4,28].

Until the previous decade, mortality rates for non-*albicans* species did not show significant changes [29]. However, *C. auris* presents unique adaptations that allow it to persist on surfaces, despite the use of common disinfectants, and to form biofilms on medical devices [30]. In addition, and apart from being resistant to many antifungals, this species is characterized by easy intra-hospital spread, especially in devices such as ventilators and catheters in patients with prolonged antibiotic treatments and immunosuppression, which facilitate its proliferation [31]. Therefore, it is crucial to study new therapeutic alternatives to stop the spread of *Candida* species and combat resistance, which is increasing in this group of yeasts.

AMPs are essential in the natural defenses of living organisms and, as such, are found in all forms of life. Due to the growing resistance of different pathogens to conventional antimicrobial treatments, AMPs have attracted notable interest as possible therapeutic tools [32]. As more AMPs are being discovered, specialized databases have been created to collect essential information related to their pharmacological use. These databases play a fundamental role in the research and development of new therapeutic strategies to combat infections [22]. Recently, the antifungal effects of different AMPs have been studied, for example, the anti-*Candida* activity of 1–18 fragment of the frog skin peptide, esculentin-1b evaluated in a *Caenorhabditis elegans* model [33], the human peptide LL-37 [34], gomesin [26], and CGA-N46 [35], among others. These, which have shown a significant reduction in the number of yeast cells at both in vitro and in vivo levels, project this group of molecules as a possible solution to fight opportunistic fungal infections that are difficult to treat. Several AMPs have also even undergone successful preclinical or clinical trials for the treatment of *Candida* infections [36].

The synthetic peptide PNR20, obtained using computational design, has shown antibacterial activity against gram-negative and gram-positive bacteria (unpublished results), as well as antifungal activity in vitro [23], with a high potential against reference strains and isolates of clinical importance, including *C. albicans* and *C. auris*. Therefore, in this study, the evaluation of the antifungal potential of PNR20 was complemented with new studies of microscopy to determine yeast-cell damage, cytotoxicity in mammalian cells and its capacity as antifungal treatment in a murine model of disseminated candidiasis caused by both species of *Candida*.

Yeasts of *C. albicans* and *C. auris* presented morphological alterations in their cellular structure, as observed through SEM (Figure 1 and Figure 2), and the exposure of these yeasts to 6.25 µM (15.25 µg/mL) of PNR20 for *C. albicans* and 50 µM (124.45 μg/mL) *C. auris* is comparable to other studies. Morphological alterations, such as to the integrity of the cell wall, induced using antifungal peptides derived from BPIFA1 were reported in *C. albicans* treated with 256 μg/mL of the peptide for 12 h [37]. Severe swelling before cell death and breakage of the outer membrane, as well as intracellular inclusion were found in *C. albicans* cells treated with 50 μg/mL of epinecidin-1 for 16 h [38]. Mo-CBP3-PepIII induced cracks and scars in the cell wall of *C. albicans* yeasts at a concentration of 2.2 μM for 24 h [39]. These reports, together with our findings highlight that PNR20 has a potent antifungal effect, given that with only 3 h of treatment, it was possible to observe effects on the cell surface and cell integrity, probably due to damage in the membrane of the treated yeasts, compared to these studies cited above where the yeasts were treated for more than 12 h.

The evaluation of cytotoxicity in the Vero cell line revealed that PNR20 induces from null to low toxicity in a concentration range that oscillates between 3.125 and 200 µM, which indicates that mammalian cell viability is not significantly compromised in the presence of PNR20 (Figure 3). These data suggest that PNR20 could be a promising option for therapeutic applications, given its favorable cytotoxicity profile in Vero cells, which agrees with the similar results for cytotoxicity of PNR20 observed in L929 cells (murine fibroblasts), which were not affected under high concentrations of this AMP [23]. However, it is important to note that additional research should be conducted to evaluate the safety and efficacy in other biological systems before considering its therapeutic use. Further studies should be also carried out to understand the functionality and pharmacodynamics of AMPs with respect to the interaction of drug–host, as well as assessing characteristics such as toxicity and easy degradation, to reduce any secondary effects in humans and add reliability to the antifungal activity [40].

The results obtained after an experimental mouse model of disseminated candidiasis, over a period of 40 days, indicated that the administration of PNR20, as daily treatment, was associated with the resolution of the infection and the survival of all mice infected with *C. albicans* or *C. auris*. It is known that systemic infection with *C. albicans* has led to 100% mortality, as occurred in our model, highlighting the virulence of the infection in the absence of treatment. Furthermore, it is of note that the group of mice treated with fluconazole, an antifungal agent commonly used in the treatment of candidiasis, showed a mortality of 70% after 29 days, indicating some efficacy but also limitations in controlling the infection (Figure 4). It is important to highlight that all the mice infected with *C. auris* and treated with PNR20 combined with fluconazole survived, possibly due to the synergistic effect of the AMP rather than the effect of the fluconazole alone, considering that the utilized isolate is resistant to fluconazole [24]. It has been already shown that the association of AMPs with antifungals can potentiate the effect of both agents against *Candida* species and are a potential therapy in the control of infections caused by resistant strains of *C. auris* [41].

After observing the in vitro therapeutic potential of PNR20, its ability to reduce the fungal load was also determined (Figure 5). The evaluation of the fungal load in the liver, spleen, and kidneys of mice infected with both *Candida* species is valid due to the tropism for certain organs that these microorganisms have in systemic infections [42]. Particularly, mice infected with *C. albicans* and *C. auris* and treated with PNR20 showed a significant decrease in the fungal load in kidneys (Figure 5). The ability of AMPs to reduce the fungal load in various studies is surprising, and it is increasingly establishing a new way of optimization and implementation of alternative therapies against resistant pathogens. A study on the use of gomesin, originating from hemocytes of the tarantula *Acanthoscurria gomesiana*, showed a significant reduction of the fungal load in mice infected with *C. albicans* with doses of 15 mg/Kg/day [26]. Likewise, another study using Jelleine-I obtained from royal jelly of *Apis mellifera* showed antifungal activity against *Candida* in vitro and in vivo [43]. Similarly, in our study, PNR20 showed efficient activity in reducing CFU of both *C. albicans* and *C. auris*. Fluconazole also showed a similar effect in reducing *C. albicans* yeasts in vitro. However, this effect was not observed when mice infected with the *C. auris* isolate were treated with fluconazole (Figure 5), due to the resistance observed with this isolate [24].

The combined effect of PNR20 and fluconazole against *Candida* was also evaluated in the BALB/c mice model, showing a promising association between this peptide and a commonly used antifungal drug, particularly in the context of the invasive candidiasis model by *C. albicans*. However, when comparing the results obtained in the *C. auris* infection model, a non-significant difference was observed between the control group and the treated group. This discrepancy can be attributed to the notable responsiveness of *C. auris* and its resistance to azoles, unlike the sensitivity shown by the *C. albicans* strain. It is essential to highlight the importance of thoroughly exploring and evaluating the possible association and potentiation between molecules such as antimicrobial and antifungal peptides. Previous studies showed the association of caspofungin and anidulafungin with AMPs [44,45], which highlights the synergism of these antifungals with AMPs and suggests the possibility of expanding the use of these molecules as alternative treatments via combined therapies. It is important to consider that changes in the size of AMPs and structural modifications can improve their activity, reduce toxicity, and are necessary to achieve optimal results and advance the effectiveness of antifungal therapies.

The histopathological analysis of the kidneys from mice infected with *C. albicans* and *C. auris* correlates with the data obtained on the fungal load and shows the protective role of PNR20 in vivo (Figure 6). An interesting finding is the possible immunomodulatory role of PNR20, as can be observed in Figure 6C, since there is a decrease in the lesions caused by *Candida* and a considerable cellular infiltrate is observed. Some AMPs may exhibit both pro- and anti-inflammatory functions. For example, they can upregulate inflammatory factors to activate the immune system, helping to eliminate invading pathogens in the early stages of an infection [46].

The results of our study suggest that PNR20 has properties that positively influence the survival of infected mice, which could be related to its ability to combat *C. albicans* and *C. auris* infection. These findings support the notion that PNR20 could be a promising candidate for the development of alternative antimicrobial therapies in the control of invasive candidiasis.

## Figures and Tables

**Figure 1 jof-09-01149-f001:**
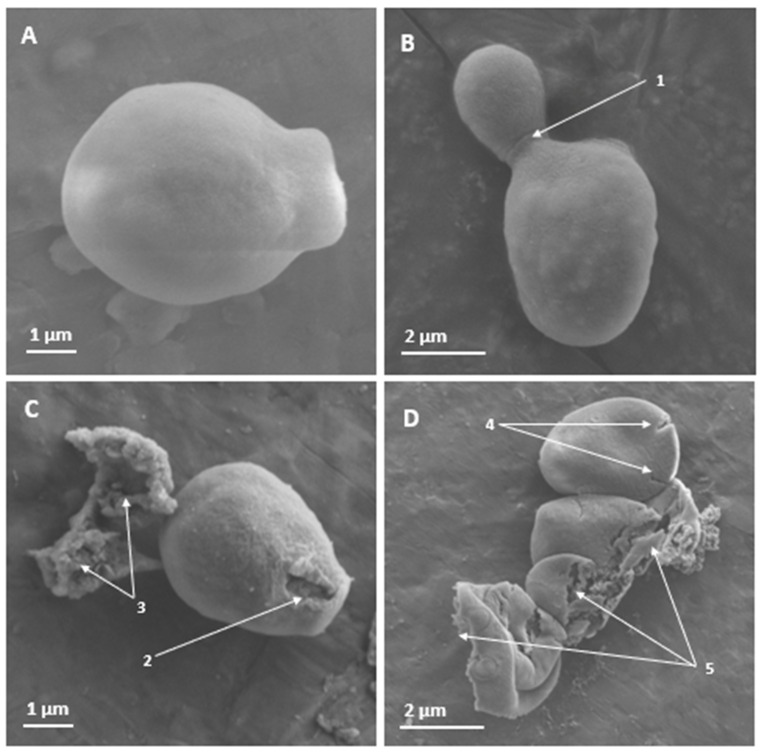
Scanning electron microscopy of the surface of *Candida albicans* ATCC 10231 yeasts treated for 3 h with PNR20, and without treatment. In non-treated yeasts (**A**,**B**), no effects are observed on the cell surface. However, apical damage is observed in yeast treated with 6.25 µM of PNR20 (**C**). With higher magnification, cracking or rupture of the cell surface can be seen on different yeasts (**D**). 1. Budding yeast. 2. Damage to the cell surface. 3. Cytoplasmic content. 4. Cracking of the cell surface. 5. Destruction of the cell surface with release of cell contents.

**Figure 2 jof-09-01149-f002:**
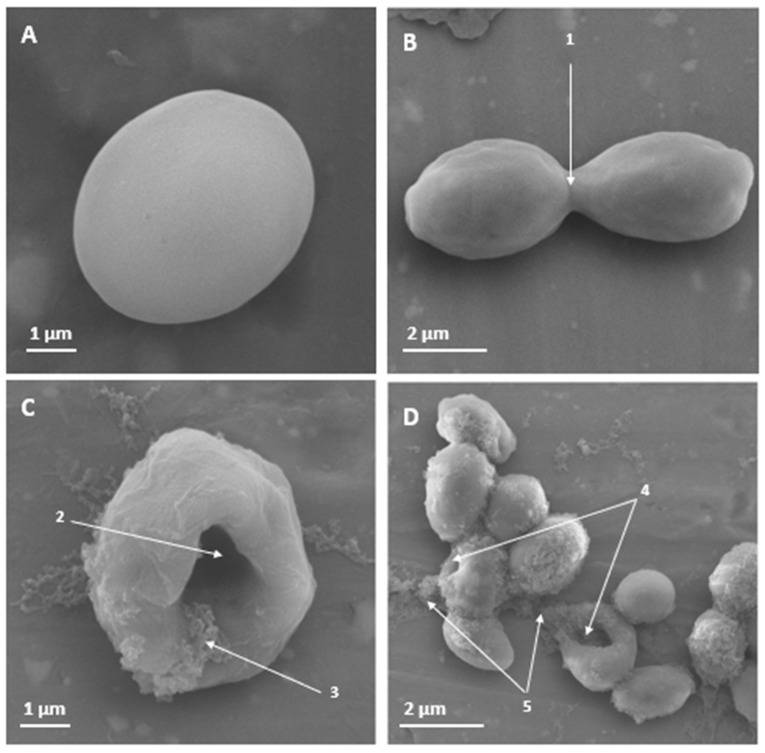
Scanning electron microscopy of *Candida auris* H0059-13-2251 yeast treated for 3 h with PNR20, and without treatment. In non-treated yeast (**A**,**B**), no effect is observed on the cell surface. However, a dent can be seen in the yeast treated with 50 µM of PNR20. With a higher magnification, a cleft is observed in the surface of the yeast that were exposed to PNR20, showing the release of the cytoplasmic content of the cells (**C**,**D**). 1. Budding yeast. 2. Damage to the cell surface. 3. Cytoplasmic contents. 4. Affectation and perturbation of the cell surface. 5. Release of cellular contents.

**Figure 3 jof-09-01149-f003:**
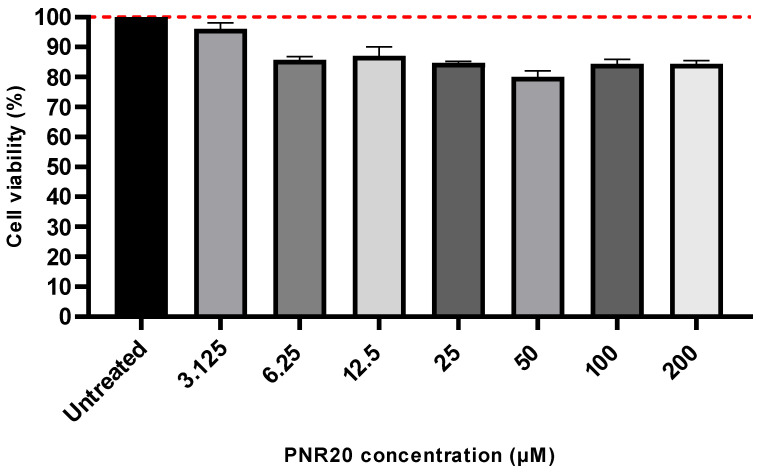
Cytotoxic of PNR20 is not induced on Vero cells. The 100% growth is indicated by the red dotted line. Peptide concentrations exposed to Vero cells ranged from 3.125 to 200 μM. Kruskal–Wallis test established that differences between treated and untreated cells were not significant (*p*-value > 0.05).

**Figure 4 jof-09-01149-f004:**
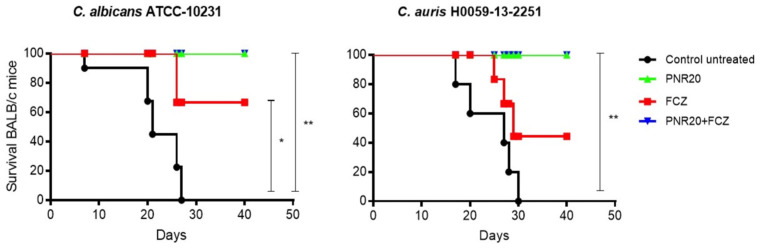
Survival curve of BALB/c mice infected intravenously with 1 × 10^3^ cells of *Candida albicans* ATCC 10231 or 1 × 10^3^ cells of *Candida auris* H0059-13-2251. All infected but untreated mice died after ~30 days after infection (black line). Mice treated with PNR20 (3 mg/Kg/day) (green line) and PNR20 combined with fluconazole (FCZ) (3 mg/Kg plus 20 mg/Kg, respectively, per day) (blue line) resolved the infection completely. Fluconazole (FCZ) treatment 20 mg/Kg/day increased surveillance of mice but did not cure the mice completely (red line). Per group, 10 mice were included. Treatment began 1 day after inoculation and continued for 7 days. Statistical significance * *p* < 0.05; ** *p* < 0.01 compared to control.

**Figure 5 jof-09-01149-f005:**
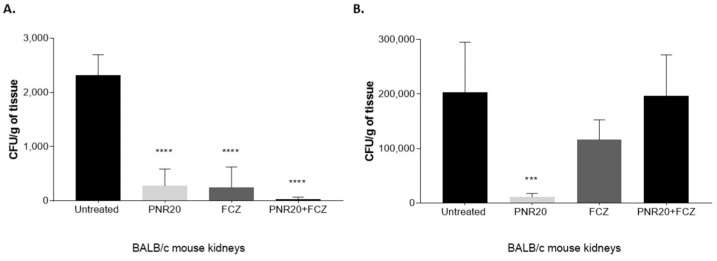
Fungal burden in kidneys of BALB/c mice infected intravenously with 1 × 10^3^ yeasts of *Candida albicans* ATCC 10231 (**A**) and 1 × 10^3^ yeasts of *Candida auris* H0059-13-2251 (**B**). Antifungal treatment was given every 24 h for 7 days after infection with each *Candida* species. Animals were treated with PNR20 (3 mg/Kg/day), with fluconazole (FCZ) (20 mg/Kg/day), or with a combined therapy (PNR20 + FCZ). As a control, infected animals were not treated but received PBS. Each bar represents the medium number of colony-forming units (CFU) per gram of tissue and the standard deviation in organs collected from six animals per group. Asterisks indicate a statistically significant difference observed between mice treated with PNR20 and/or fluconazole and infected but untreated mice. One way-ANOVA Statistical significance *** *p* < 0.001; **** *p* < 0.0001.

**Figure 6 jof-09-01149-f006:**
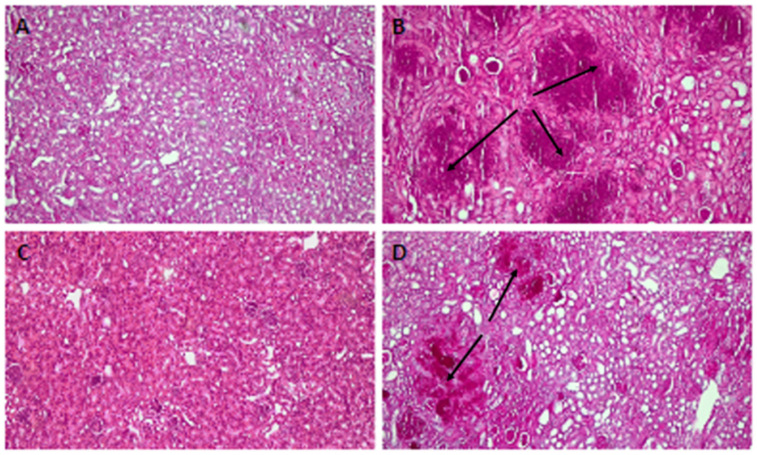
Histopathology of kidneys from BALB/c mice infected intravenously with 1 × 10^3^ cells of *Candida auris* H0059-13-2251. (**A**) Tissue from uninfected mice. (**B**) Infected untreated control. (**C**) Tissue from infected animals treated with 3 mg/Kg/day of PNR20 for 7 days, showing preserved structure and absence of lesions and fungal cells; observe the intense cellular infiltration. (**D**) Kidneys from mice infected and treated with 20 mg/Kg/day of fluconazole for 7 days. Arrows indicate yeast concentrates.

## Data Availability

Data are contained within the article.

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
