# Peer review of "Therapeutic Use of the Antimicrobial Peptide PNR20 to Resolve Disseminated Candidiasis in a Murine Model"

_jof, 2023, doi:10.3390/jof9121149_

Round 1

Reviewer 1 Report

Comments and Suggestions for Authors

This manuscript is a follow-up study on the in vitro activity of the peptide PNR20 against C. albicans and C. auris, where the in vivo activity is studied. It is generally well written and clear, however some comments for improvement:

1. In the materials and methods it is not clear how the peptide was synthesised. If this was done as in reference 23, please indicate this

2. For many of the results of the SEM analyses, including figure legends, the authors state that he cell membrane was damaged. While this may be true, the images show the cell surface, i.e. the cell wall, to be damaged. Since these images do not differentiate between the cell membrane and the cell wall, I suggest the authors be more careful in their use of the terms membrane/wall as these are distinct structures.

3. The authors only provide images of the C. auris infection of the kidneys, however they state in the materials and methods that many other organs were assessed. Why are these images not provided? In addition, it may be of interest to also see the morphology of C. albicans in the organs. If hyphae are formed, it may also explain the differences in the cfus as hyphae may be damaged during homogenisation, leading to lower cfus than expected? 

Minor errors

Page 3, section 2.3, line 128: established is spelled incorrectly

Page 4, section 2.5, line 164: Format of reference (Rossi et al., 2012) is incorrect

Page 6, ln 208-210: Please revise the grammar of this section

Comments on the Quality of English Language

As indicated -some minor corrections

Author Response

Thanks to the reviewer for the time to read and evaluate the manuscript and for the valuable comments and suggestions that improve our study.

  1. In the materials and methods it is not clear how the peptide was synthesized. If this was done as in reference 23, please indicate this

A/: Yes, the peptide PNR20 was synthesized, commercially, by the company Peptide 2.0, as indicated in the reference 23. This is now described in section 2.1. of the manuscript.

  1. For many of the results of the SEM analyses, including figure legends, the authors state that the cell membrane was damaged. While this may be true, the images show the cell surface, i.e. the cell wall, to be damaged. Since these images do not differentiate between the cell membrane and the cell wall, I suggest the authors be more careful in their use of the terms membrane/wall as these are distinct structures.

A/: Thanks to the reviewer for the valuable observation. We agree with being more careful with the use of the terms wall/membrane, as such, we now use the term “cell surface” through the manuscript to indicate that probably the cell membrane is damaged, but that the images are not sufficient to distinguish between cell wall or cell membrane.

  1. The authors only provide images of the C. auris infection of the kidneys, however they state in the materials and methods that many other organs were assessed. Why are these images not provided? In addition, it may be of interest to also see the morphology of C. albicans in the organs. If hyphae are formed, it may also explain the differences in the cfus as hyphae may be damaged during homogenisation, leading to lower cfus than expected?

A/: As the reviewer noticed, spleens and livers were also collected. However, the fungal burden in these organs was very low, therefore, results were not presented and histological analyses were not done. This is now mentioned in the materials and methods section, as well as in the results section.

Regarding the images from C. albicans infection, we mentioned in the manuscript that “similar results were observed in the kidneys of mice infected with C. albicans”, considering that no hyphae or pseudohyphae formation was observed (this is now stated in the ms). Therefore, the difference in number of CFU in kidneys is not due to damage caused during homogenization, but perhaps to the capability of C. auris to proliferate faster and greater that C. albicans, as it has been reported in mouse and porcine skin models.

Minor errors

Page 3, section 2.3, line 128: established is spelled incorrectly

A/: Corrected in the manuscript

Page 4, section 2.5, line 164: Format of reference (Rossi et al., 2012) is incorrect

A/: This was corrected in the manuscript and the order of all references was modified accordingly.

Page 6, ln 208-210: Please revise the grammar of this section

A/: The grammar was revised in this section.

Reviewer 2 Report

Comments and Suggestions for Authors

I have read the paper entitled"Therapeutic use of the antimicrobial peptide PNR20 to resolve disseminated candidiasis in a murine model". I think the idea is good, appropiate methodologies were applied and it is well writte. However, I have some  comments and questions and I think the authors should explain certain facts of the work.

Comments: Introduction is very long, it should be shortened and written as necessary to justify the importance of the study, The discussion is also very long, it should be shortened and take into account only the similarities or discrepancies of the results of this study with others already published. I have some doubts about the murine model:  Why did you use such a low inoculum of the yeasts of the 2 species to establish the infection? In therapeutic efficacy models it is advisable to use larger inocula precisely to test the efficacy of the drug being tested. You not specify the route of administration of fluconazole. Why you use only 1 dose of the peptide (3mg/kg/day) in the treatment scheme? I insist in the therapeutic efficacy studies, it is advisable to try several doses, the importat thing is to find the lowest doses that is effective. in the fungal load study of the murine model used in this work, you gave treatment 7 days and the animals were sacrificed on day 8 after infection. Why did not study a group of animals that were sacrificed some time later (for example, day 20) and thus see the potency of the peptide, long after having administered it?. Figure 1 and 2: You say that the damage is in the plasma membrane. I think the damage include also cell wall.

Author Response

Thanks to the reviewer for the time to read and evaluate the manuscript and for the valuable comments and suggestions that improve our study.

Comments: Introduction is very long, it should be shortened and written as necessary to justify the importance of the study, The discussion is also very long, it should be shortened and take into account only the similarities or discrepancies of the results of this study with others already published.

A/: We appreciate the comment of the reviewer, however all the authors consider that both the introduction and discussion contain relevant information in the context of the experiments carried out. As such, these sections shouldn’t be shortened.

I have some doubts about the murine model:  Why did you use such a low inoculum of the yeasts of the 2 species to establish the infection? In therapeutic efficacy models it is advisable to use larger inocula precisely to test the efficacy of the drug being tested.

A/: We agree with the reviewer's comments that in certain studies higher inoculums are used to establish infection. However, in our model, particularly, the inoculum was low because the BALB/c mice were immunosuppressed with cyclophosphamide (as specified in section 2.5). Cyclophosphamide is a powerful immunosuppressant and if we increase the inoculum, it can cause the death of the animals even a few days after infection with either C. albicans and C. auris due to their recognized pathogenicity. Infection with 10^3 yeasts, in immunosuppressed mice has been reported elsewhere (Rossi et al 2012).

You not specify the route of administration of fluconazole.

A/: This information is now included in the manuscript in section 2.5

Why you use only 1 dose of the peptide (3mg/kg/day) in the treatment scheme?

B/: Thanks to the reviewer for noticing this. Animals we actually treated for 7 days. This information is now included in the manuscript in section 2.5

I insist in the therapeutic efficacy studies, it is advisable to try several doses, the important thing is to find the lowest doses that is effective

A/: The dose of the peptide used for treatment, which was calculated based on the molecular weight and the MIC of the peptide, as established previously [Torres et al 2023], was chosen considering the results of the study by Rossi et al. 2012. This is now mentioned in the manuscript, section 2.5. In addition, funding was insufficient to carry out additional experiments with mice.

In the fungal load study of the murine model used in this work, you gave treatment 7 days and the animals were sacrificed on day 8 after infection. Why did not study a group of animals that were sacrificed some time later (for example, day 20) and thus see the potency of the peptide, long after having administered it?

A:/ In our study we did leave some animals for 40 days after infection, in order to evaluate the potency/efficacy of the peptide, as it is stated in the manuscript (surveillance assays). With this experiment we could determine the effectiveness of the treatment, given that a significant decrease in mortality of animals treated with the peptide PNR20 was observed, compared to infected but untreated animals.

Figure 1 and 2: You say that the damage is in the plasma membrane. I think the damage include also cell wall.

A./ As it was suggested by the first reviewer, we agree with being more careful with the use of the terms wall/membrane. Therefore, we now use the term “cell surface” through the manuscript to indicate that probably the cell membrane is damaged, but that the images are not sufficient to distinguish between cell wall or cell membrane.